# The Hydrochemistry, Ionic Source, and Chemical Weathering of a Tributary in the Three Gorges Reservoir

**Qianzhu Zhang [1,\*], Ke Jin [1], Linyao Dong [2], Ruiyi Zhao [3,4], Wenxiang Liu [1], Yang Lu [1], Xiaoqing Gan [1], Yue Hu [1] and Cha Zhao [1]**

[1]  Chongqing Branch Institute, Changjiang River Scientific Research Institute, Chongqing 400026, China
[2]  Soil and Water Conservation Department, Changjiang River Scientific Research Institute, Wuhan 430000, China
[3]  College of Architecture and Urban Planning, Chongqing Jiaotong University, Chongqing 400074, China
[4]  Chongqing Key Laboratory of Karst Environment, School of Geographical Sciences, Southwest University, Chongqing 400715, China
[\*]  Correspondence: qianzhuzhang@163.com; Tel.: +86-23-86815647; Fax: +86-23-88192718

**Abstract:** Riverine dissolved matter reflects geochemical genesis information, which is vital to understand and manage the water environment in a basin. The Ganjing River located in the hinterland of the Three Gorges Reservoir was systematically investigated to analyze the composition and spatial variation of riverine ions, probe the source and influencing factors, and assess the chemical weathering rates and $CO_2$ consumption. The results showed that the total dissolved solid value ($473.31 \pm 154.87$ mg/L) with the type of "$HCO_3^- - Ca^{2+}$" was higher than that of the global rivers' average. The hydrochemical parameters were relatively stable in the lower reservoir area of the Ganjing River, which was largely influenced by the backwater of Three Gorges Reservoir. The carbonate weathering source contributed 69.63% of TDS (Total dissolved solids), which generally dominated the hydrochemical characteristics. The contribution rates of atmospheric rainfall were relatively low and stable in the basin, with an average of $4.01 \pm 1.28\%$. The average contribution rate of anthropogenic activities was 12.05% in the basin and even up to 27.80% in the lower reservoir area of the Ganjing River, which illustrated that the impoundment of Three Gorges Reservoir had brought great challenges to the water environment in the reservoir bay. The empirical power functions were tentatively proposed to eliminate the dilution effect caused by runoff discharge on the basis of previous studies. Accordingly, the rock weathering rate was calculated as 23.54 t/km$^2$ in the Ganjing River Basin, which consumed atmospheric $CO_2$ with a flux of $6.88 \times 10^5$ mol/y/km$^2$. These results highlight the geochemical information of tributaries in the hinterland of the Three Gorges Reservoir, have significant implications for understanding the impact of impoundment, and provide data support for the integrated management of water resources in the Ganjing River Basin.

**Keywords:** hydrochemistry; source apportionment; weathering rate; $CO_2$ consumption

## 1. Introduction

River material transport and its ecological environment effects are directly related to the evolution of estuarine and coastal marine ecosystems [1,2]. Furthermore, through analyzing riverine ionic composition and sources, the information about the geomorphic evolution, hydrological process, bioecological process, and human activities in a basin can be obtained [3–7]. Since 1970, researchers have conducted extensive studies on the chemical characteristics and their controlling factors in major global rivers. Most results showed that the riverine ionic composition mainly came from rock weathering, and the contribution of evaporate dissolution, atmospheric rainfall and human activities should also not be neglected [6,8–11]. In addition, the chemical weathering process can absorb atmospheric $CO_2$ and convert it into $HCO_3^-$, which is usually transported into oceans via rivers and

deposited in the form of carbonates. The above process is also an important global carbon cycle, which has a vital impact on climate change on a geological time scale [12,13].

At present, the global environment is facing huge risks and challenges [14], and human activities are widely affecting the surface environment. According to the latest statistics of the International Commission on Large Dams, there are 58,000 dams with a height of more than 15 m or a reservoir capacity of more than $3 \times 10^8$ m$^3$ in the world (http://www.icold-cigb.org (accessed on 1 October 2022)). The impoundment of global reservoirs has destroyed connectivity and hydrological conditions, hindered the migration of biogenic elements along the river network, and affected the ecological environment in downstream areas [2,15–17]. The Three Gorges Reservoir (TGR) is located in Chongqing and Hubei in southwest China, regulates 56% of the Yangtze River, and plays a very important strategic role in promoting the economic exchanges between Eastern and Western regions of China [18]. Since the full impoundment in 2010, TGR and its tributaries have suffered from hydrological process changing, water level rising, and flow velocity slowing down [19–21]. According to the investigation, about 38 tributaries have displayed eutrophication and experienced water blooms in varying degrees. The water quality in the tributary estuary is faced with a more serious situation compared with that in the TGR [22]. However, the impacts of impoundment on water quality of TGR tributaries remain poorly understood.

Located in the hinterland of TGR, the Ganjing River is a vital tributary to support the economic and social development in Zhongxian County. About 8 km of backwater area is formed due to the impoundment of TGR, which affects the water quality and hydrology in the Ganjing River Basin. In fact, the water deterioration, eutrophication, and algal blooms are seriously restricting the ecological security of the basin. However, the spatial characteristics of hydrochemistry and its controlling factors remains unclear. According to the Fourteenth Five Year Plan for Ecological Environment Protection in Chongqing, an ecological regulation dam will be built in the Ganjing River, which largely brings unknown impacts. Before that, it is necessary to fully understand the water environment. Based on an investigation carried out in the whole basin, this study aims to (1) investigate the ionic concentrations and spatial variation, (2) study the ionic source and its influencing factors, and (3) assess the chemical weathering rates and $CO_2$ consumption in the basin, which are significant to grasp the evolution process of the water environment in a large-scale reservoir ecosystem and provide theoretical support for ecological regulation dam construction.

## 2. Study Area and Methods

### 2.1. Study Area

The Ganjing River is located at the north of the Yangtze River, with the range of 107°40′01″~108°04′00″ E and 30°15′29″~30°39′06″ N. The basin is composed of three tributaries, the Baishi River, Sanhui River, and Huangjin River, with a whole area of 910 km$^2$ (Figure 1). Due to the construction and impoundment of TGR, the backwater zone is formed along the lower reach of the Ganjing River, which is farthest from the estuary of the Huangjin River. The subtropical humid monsoon climate predominates, with an average annual temperature of 18.1 °C and annual precipitation of 1200 mm. The average annual discharge is $4.82 \times 10^8$ m$^3$, 70% of which happens in the flood season (from May to September). The river flows through the parallel ridge valley area on the eastern edge of the Sichuan Basin, with an elevation of 700~1000 m in the low mountains and hills. The Middle Triassic (T$_2$) of the Mesozoic strata, including argillaceous limestone, dolomitic limestone, shale, and mudstone, to the upper Jurassic Suining Formation (J$_3$s) and Penglaizhen Formation (J$_3$p) sandstone and mudstone are exposed in the basin. Several towns with large populations, such as Shihuang, Jinji, Sanhui, Baishi, Jinjin, and Zhongzhou, are successively located along the river. The agricultural cultivation area accounts for more than 60% of the total area, where grain crops (rice, wheat, soybean, and potato) and economic crops (citrus, Chinese herbal medicine, and sericulture) are widely planted in the basin.

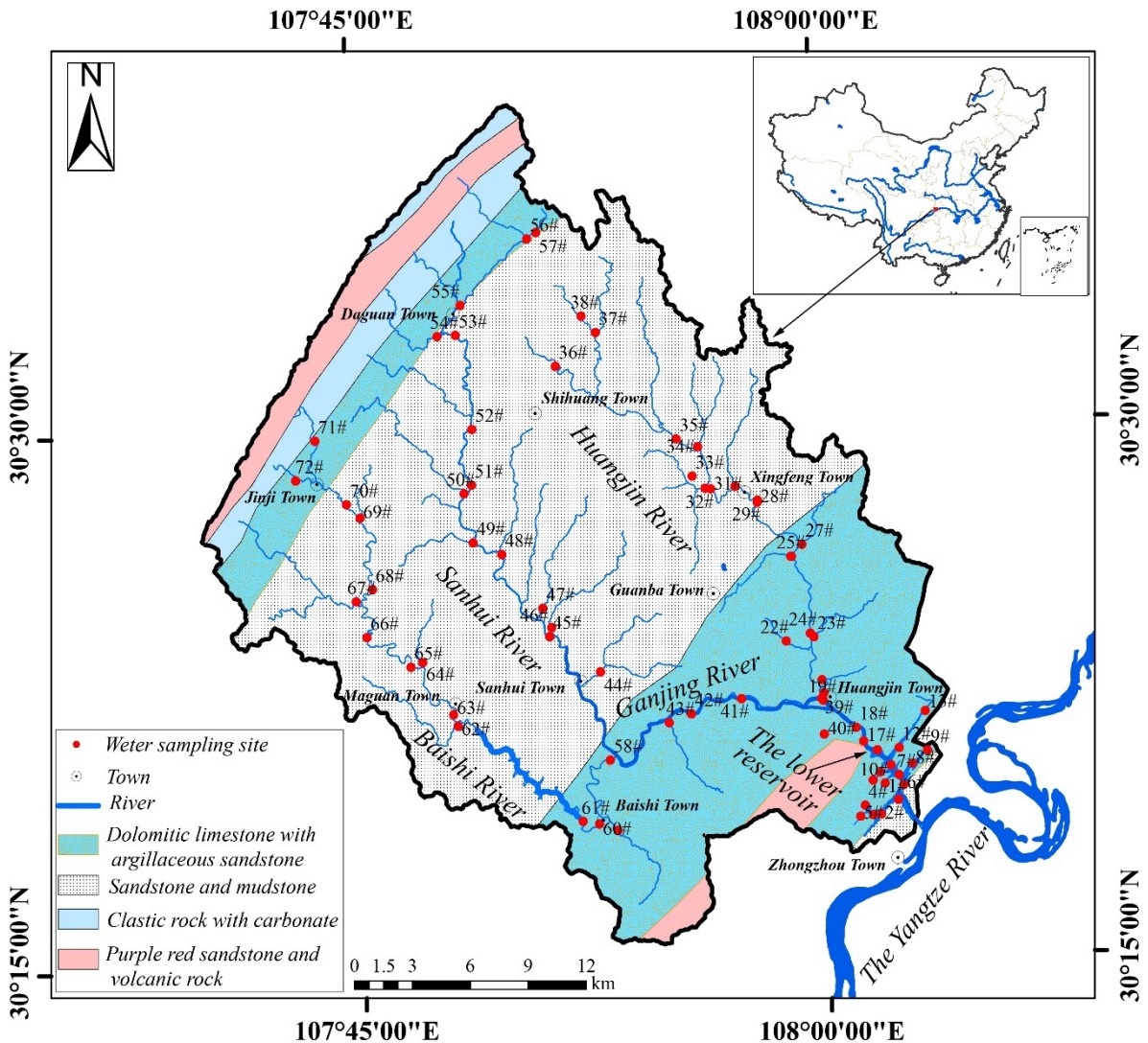

**Figure 1.** The lithology and sampling sites in the Ganjing River Basin.

### 2.2. Sampling

The water environment was systematically monitored from 24–29 December 2020 along the main tributaries of the Ganjing River. At the same time, 72 samples were collected, of which 19 (1#~19#) samples were distributed in the lower reservoir area of the Ganjing River, 19 samples (20#~38#) in the Huangjin River, 19 samples (39#~57#) in the Sanhui River, and 15 samples (58#~72#) in the Baishi River. The water temperature (Temp), pH, dissolved oxygen (DO), conductivity (Cond), total dissolved solid (TDS), oxidation–reduction potential (ORP), and resistivity (RES) were measured by AP-7000, Aquaread Co., England, in situ, with the accuracy of $\pm0.1$ °C, $\pm0.01$, 0.01 mg/L, $\pm1$ µS/cm, $\pm0.1$ mg/L, 0.1 mv, and $\pm0.01$, respectively. The sampling apparatus was washed three times with the river water in situ before water samples were collected. The water samples were collected at about 1 m below the water surface and sealed well for preservation.

### 2.3. Methodology

The water samples were filtered through Whatman GF/F filters (pore size 0.45 µm), and then the filtrate was divided into two portions. One was used for measuring major cations and dissolved silicon (DSi), and the other for analyzing major anions. The remaining samples were used to titrate the total alkalinity. The filtered sample was placed in a refrigerator at 5 °C for testing, and all experiments were completed in one week. $K^+$,

$Na^+$, $Ca^{2+}$, $Mg^{2+}$, and DSi were analyzed by an ICP-OES plasma spectrometer (ICAP 7000 plus). $Cl^-$, $NO_3^-$, and $SO_4^{2-}$ were tested by Dionex ICS-600 ion chromatography. The test accuracy was above 0.5%. Total alkalinity was titrated by a Metrohm 848 Titrino plus automatic potentiometric titrator (produced by HACH, Loveland, CO, USA). The calculation was conducted through the Gran graphic method, and the average value of three calculations was selected. When pH values in the water ranged from 6 to 10, the carbonic acid system in the water was dominated by $HCO_3^-$.

## 3. Results

### 3.1. Physical and Chemical Parameters

During the sampling period, the pH value ranged from 7.87 to 8.64, showing moderately alkalescent. The TDS value fluctuated from 213.00 to 994.33 mg/L, with an average of $473.31 \pm 154.87$ mg/L, which was higher than the global rivers' average [11]. It was also higher than that in the Xijiang River Basin, where carbonate rock was widely distributed (150.60~212.70 mg/L) [23], and about twice the TDS of the Yangtze River (231~261 mg/L), Wujiang River (273 mg/L), and Jialing River (269 mg/L) in Chongqing [6]. The above phenomenon was likely due to the sampling period in December with little rainfall when the dissolved solute was relatively enriched in the river. The riverine DO, Cond, and ORP values were 8.7~12.77 mg/L, 328.67~1530.33 μS/cm, and 45.47~198.20 MV in the basin, respectively. The riverine hydrochemical parameters showed obvious spatial distribution characteristics. Specifically, the average of pH, DO, Cond, and TDS in the downstream of the Ganjing River were lower than those in the Huangjin River, Sanhui River, and Baishi River (Figure 2). Meanwhile, the spatial variation of riverine temperature and ORP was reversed. In addition, the standard deviations of water temperature, pH, DO, Cond, and TDS in the lower reservoir area of the Ganjing River were smaller than those in other river sections. The above analysis results illustrated that hydrochemical parameters were relatively stable in the lower reservoir area owing to the backwater caused by TGR.

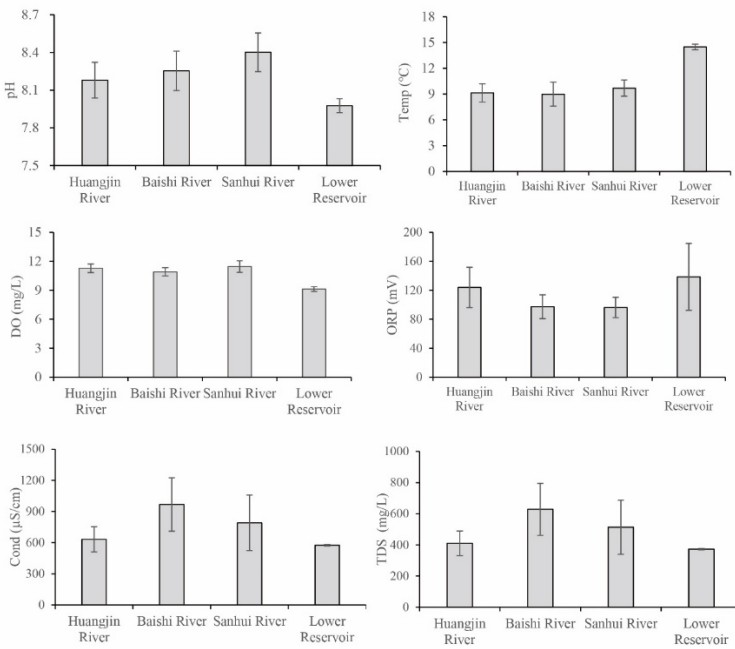

**Figure 2.** The physical and chemical parameters in the different tributaries' sections.

### 3.2. The Analysis of Riverine Main Ions

The average concentrations of $K^+$, $Na^+$, $Ca^{2+}$, and $Mg^{2+}$ were $61.15 \pm 25.51$ μmol/L, $599.44 \pm 245.14$ μmol/L, $1830.37 \pm 783.49$ μmol/L, and $545.66 \pm 312.49$ μmol/L, respectively. $Ca^{2+}$ was the main component in the cations, accounting for 43.83~74.53% (Figure 3). The $HCO_3^-$ ranged from 1542.87 to 5542.16 μmol/L and accounted for 70.48% of anions.

Concentrations of $Cl^-$, $NO_3^-$, and $SO_4^{2-}$ were from 97.53 to 743.68 μmol/L, 15.77 to 464.01 μmol/L, and 164.43 to 4943.44 μmol/L, respectively. The concentrations of $Ca^{2+}$ and $HCO_3^-$ in the lower reservoir area were lower than those in other river sections, which was likely related to supersaturated deposition of calcium carbonate caused by $HCO_3^-$ absorption of aquatic organisms or $CO_2$ release because of reduced hydraulic conditions. Compared with the main tributaries in the Yangtze River Basin, the ion composition in the Ganjing River was similar to that of the Wujiang River, Jialing River, and Minjiang River Basin [6].

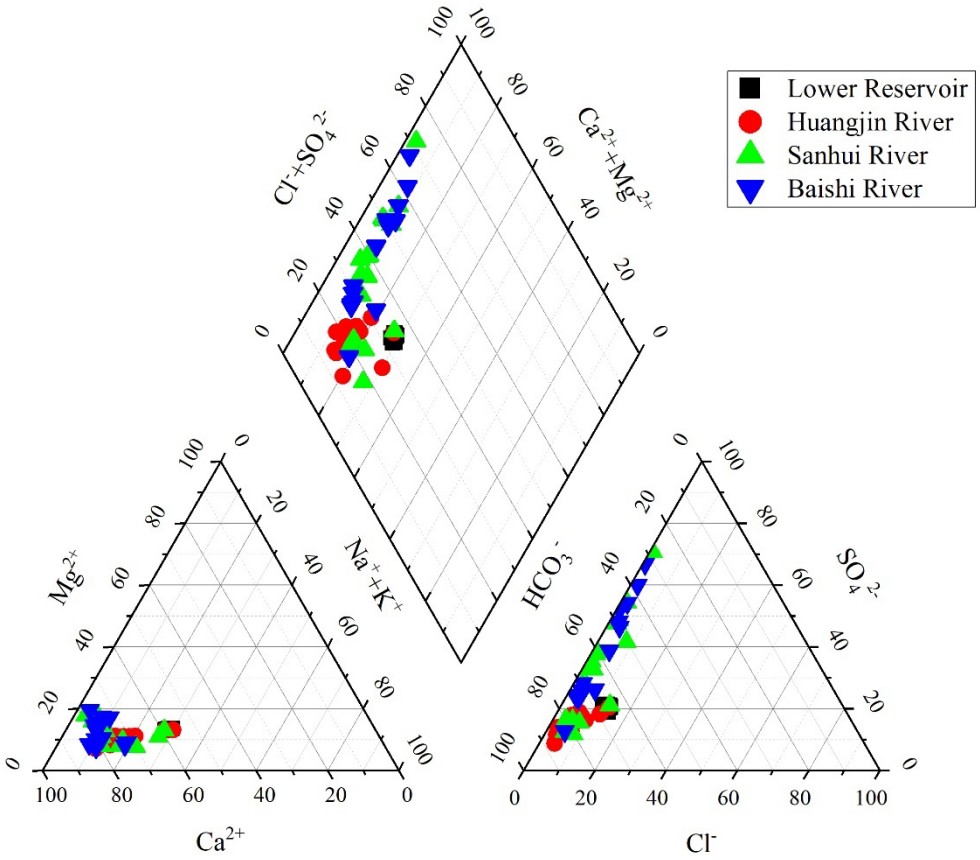

**Figure 3.** Piper diagram for the water samples in the Ganjing River Basin.

The main cationic charges ($TZ^+ = K^+ + Na^+ + Ca^{2+} + Mg^{2+}$) and the anionic charges ($TZ^- = Cl^- + NO_3^- + SO_4^{2-} + PO_4^{3-} + HCO_3^-$) ranged from 2323.06 to 13,414.87 ueql/L and 2214.05 to 13,146.80 ueql/L, with the average of positive and negative charges 5412.59 ± 2019.10 ueql/L and 5335.50 ± 2018.89 ueql/L, respectively. The inorganic charge balance indexes were mostly within ±10%, revealing that the water was less affected by anthropogenic pollution. The ionic charges had significant spatial variation, with the average in the Baishi River, Sanhui River, Huangjin River, and the lower reservoir area of the Ganjing River being 7130.20 ueql/L, 5988.30 ueql/L, 4606.00 ueql/L, and 4287.49 ueql/L, respectively.

### 3.3. Correlation Analysis of Hydrochemical Parameters

Riverine pH and ORP were significantly correlated with DO, Cond, RES, and TDS (Table 1), suggesting that pH and ORP were likely the basic indices of hydrochemical parameters. The $Ca^{2+}$ and $Mg^{2+}$ significantly correlated with TDS, with correlation coefficients as high as 0.96 and 0.87, which revealed that $Ca^{2+}$ and $Mg^{2+}$ occupied an absolute advantage in the total ions. Among the main anions, $SO_4^{2-}$ and $HCO_3^-$ were significantly correlated with TDS values and constituted the main components, accounting for 88.9% of the total anions. There was a very significant positive correlation between $Na^+$ and $Cl^-$

(R = 0.96, $p < 0.01$), which were authenticated as the main components of substances from human activities. In addition, the TDS value had a significant negative correlation with $Na^+$ and $Cl^-$, which revealed that the above anthropogenic effects mainly occur in areas with relatively low TDS. A significant linear correlation ($p < 0.01$) existed between $HCO_3^-$ and $K^+$, and $Ca^{2+}$ and $Mg^{2+}$, which showed a similar source and migration relationship among ions.

**Table 1.** The correlation analysis among the hydrochemical indexes.

| | pH | ORP | DO | EC | RES | TDS | $K^+$ | $Na^+$ | $Ca^{2+}$ | $Mg^{2+}$ | $Cl^-$ | $NO_3^-$ | $SO_4^{2-}$ | DSi | $HCO_3^-$ |
|---|---|---|---|---|---|---|---|---|---|---|---|---|---|---|---|
| pH | 1 | | | | | | | | | | | | | | |
| ORP | −0.435 ** | 1 | | | | | | | | | | | | | |
| DO | 0.757 ** | −0.383 ** | 1 | | | | | | | | | | | | |
| EC | 0.534 ** | −0.380 ** | 0.227 | 1 | | | | | | | | | | | |
| RES | −0.302 ** | 0.254 * | 0.088 | −0.853 ** | 1 | | | | | | | | | | |
| TDS | 0.534 ** | −0.380 ** | 0.227 | 1.00 ** | −0.853 ** | 1 | | | | | | | | | |
| $K^+$ | −0.042 | −0.233 * | −0.248 * | 0.454 ** | −0.478 ** | 0.454 ** | 1 | | | | | | | | |
| $Na^+$ | −0.662 ** | 0.416 ** | −0.721 ** | −0.445 ** | 0.178 | −0.445 ** | 0.149 | 1 | | | | | | | |
| $Ca^{2+}$ | 0.594 ** | −0.367 ** | 0.340 ** | 0.964 ** | −0.790 ** | 0.964 ** | 0.308 ** | −0.571 ** | 1 | | | | | | |
| $Mg^{2+}$ | 0.430 ** | −0.275 * | 0.057 | 0.866 ** | −0.703 ** | 0.866 ** | 0.262 * | −0.347 ** | 0.858 ** | 1 | | | | | |
| $Cl^-$ | −0.654 ** | 0.394 ** | −0.783 ** | −0.392 ** | 0.094 | −0.392 ** | 0.194 | 0.959 ** | −0.522 ** | −0.291 * | 1 | | | | |
| $NO_3^-$ | −0.277 * | 0.027 | −0.094 | −0.091 | 0.070 | −0.091 | −0.198 | 0.093 | −0.118 | −0.071 | 0.093 | 1 | | | |
| $SO_4^{2-}$ | 0.121 | −0.271 | 0.125 | 0.514 ** | −0.281 | 0.514 ** | −0.115 | −0.248 | 0.369 * | −0.140 | −0.230 | 0.152 | 1 | | |
| DSi | −0.159 | 0.086 | −0.324 ** | −0.032 | 0.058 | −0.032 | 0.118 | 0.082 | −0.061 | 0.127 | 0.067 | −0.190 | −0.118 | 1 | |
| $HCO_3^-$ | 0.435 ** | −0.262 * | 0.372 ** | 0.590 ** | −0.611 ** | 0.590 ** | 0.314 ** | −0.376 ** | 0.626 ** | 0.248 * | −0.428 ** | −0.124 | 0.365 * | −0.153 | 1 |

** Correlation is significant at the 0.01 level; *correlation is significant at the 0.05 level.

## 4. Discussions

### 4.1. Spatial Differentiation of Hydrochemical Characteristics

4.1.1. Physical and Chemical Parameters Variation

The water temperature gradually rose from upstream to downstream in the lower reservoir area of the Ganjing River, which was significantly higher than that in the tributaries such as the Huangjin River, Ganjing River, and Baishi River (Figure 4). Meanwhile, the temperature difference between the main stream and tributaries in the reservoir area was not obvious. The above phenomena were likely due to the water mixed effect and stratification in the lower reservoir. The pH value in the Sanhui River was generally higher than that of other river sections, and that was relatively low in the reservoir area. Due to the confluence of tributaries along the river, the pH value in the main stream of the Huangjin River, Sanhui River, and Baishi River decreased with fluctuation. The riverine ORP and DO values fluctuated with no special change pattern, which was likely affected by many factors such as velocity, water depth, aquatic organism absorption, and pollutant degradation. However, it was worth mentioning that the DO values in the reservoir area were generally lower than those of other river sections, which further confirmed the impact of limited water temperature exchange conditions owing to the large depth in the reservoir. The riverine TDS values had significant differences in different regions of the basin, especially in main streams. The TDS values in the main stream of the Baishi River were generally higher than those of other rivers, with a gradually decreasing trend from upstream to downstream. The TDS values in the Sanhui River were high at the midstream and low at both the upstream and downstream, which were consistent with its tributaries. The TDS values in the main stream of the Huangjin River gradually decreased and were generally lower than those in the Sanhui River and Baishi River. The TDS values in the lower reservoir area of the Ganjing River were generally lower than that in the other three main tributaries' basins and were relatively stable as a whole with little spatial difference. Overall, the TDS values in the main streams generally showed a downward trend from upstream to downstream, as a result of the inflow of tributaries.

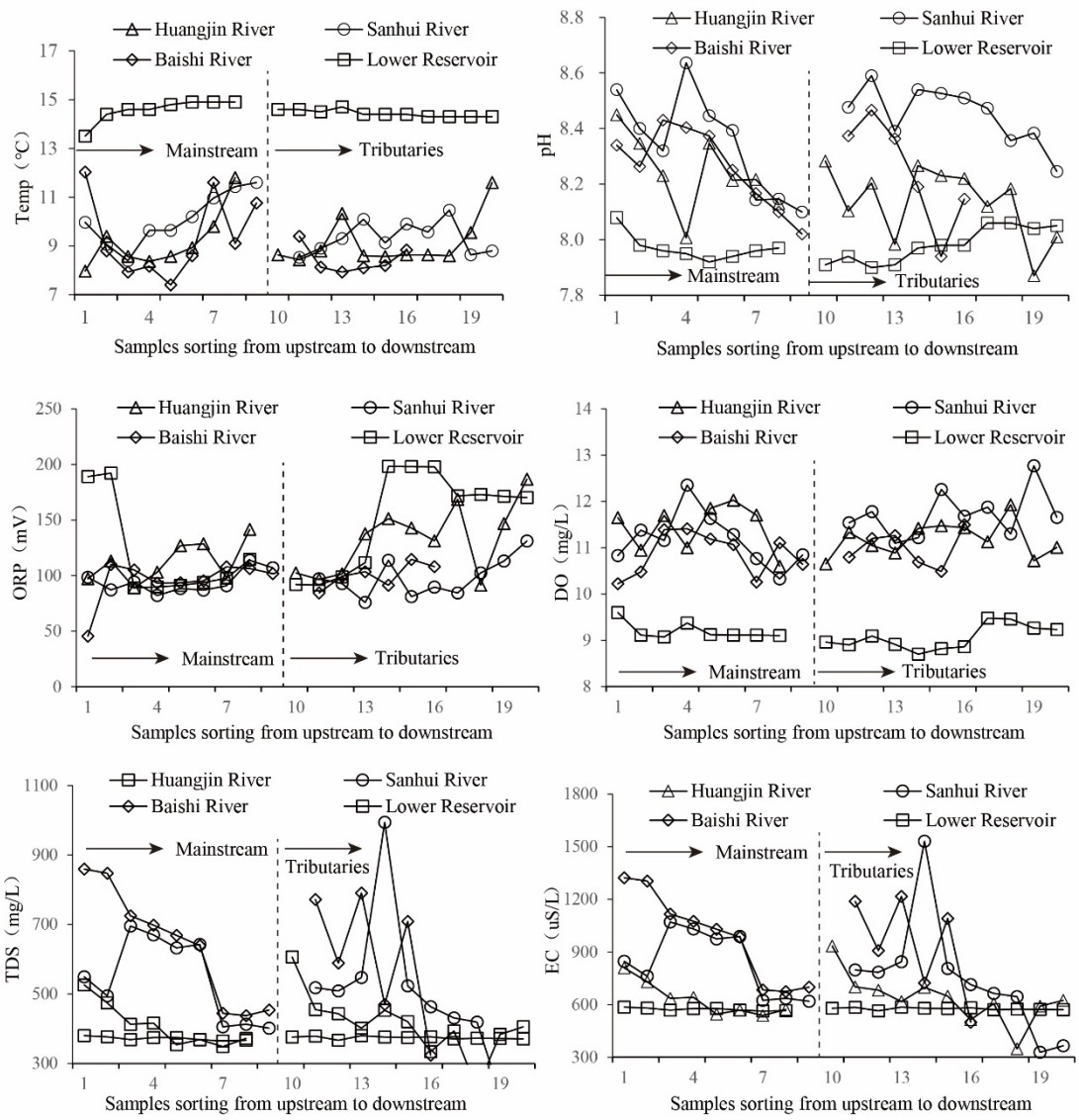

**Figure 4.** The spatial variation of physicochemical indexes in the Ganjing River Basin.

### 4.1.2. Spatial Characteristics of Cations

According to the spatial variation of main cations in different regions, the concentration of $K^+$, $Na^+$, $Ca^{2+}$, and $Mg^{2+}$ were relatively stable in the lower reservoir area of the Ganjing River (Figure 5), and they had different spatial variation laws, respectively. The $K^+$ concentrations were generally higher in the Baishi River than those in other regions and generally decreased from upstream to downstream in the main stream of the Baishi River and Huangjin River. However, the $K^+$ concentrations increased gradually from upstream to downstream in the main stream of the Sanhui River, and those were relatively stable as a whole in the reservoir area. The $Na^+$ concentrations had significant spatial variations in different regions, which was the most in the main stream. In addition, the $Na^+$ concentrations were generally higher in the reservoir area than those in other regions, followed by the Huangjin River. The variations of $Na^+$ concentrations appeared to cross-distribute in the main streams of the Baishi River and Sanhui River, which were higher in the upper reaches of the Baishi River than those in the lower reaches of the Sanhui River. The $Na^+$ concentrations in the main streams of the Huangjin River and Baishi River decreased gradually, while those in the main stream of the Ganjing River increased first and then decreased. The $Na^+$ concentration in the estuary of the Huangjin River and Sanhui River rapidly increased to the values in the reservoir, which was largely affected by the backflow

of the reservoir area. The $Ca^{2+}$ and $Mg^{2+}$ concentrations had similar spatial variations in the basin. Except for individual points in the upper reaches of the Baishi River and Sanhui River, the $Ca^{2+}$ and $Mg^{2+}$ concentrations in the Huangjin River, Baishi River, and Sanhui River gradually decreased from upstream to downstream, and those in the reservoir area were generally stable.

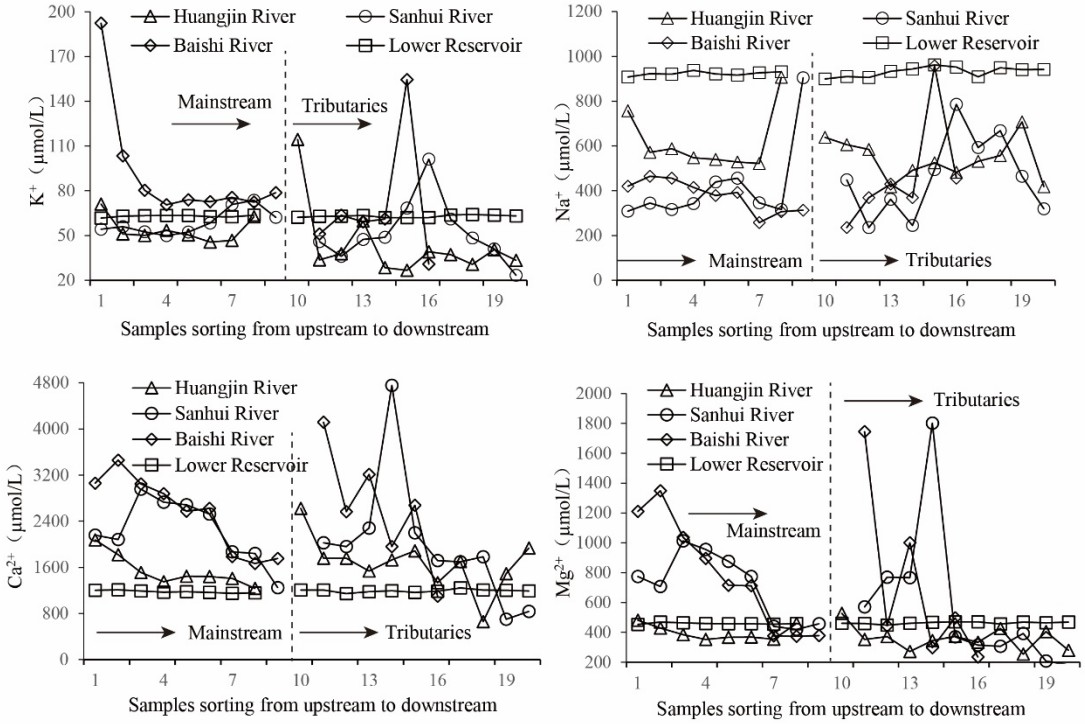

**Figure 5.** The spatial variation of major cations in the Ganjing River Basin.

### 4.1.3. Spatial Characteristics of Anions

The $Cl^-$ concentrations decreased gradually from upstream to downstream as a whole in the main streams of the Huangjin River and Baishi River, except for individual points (Figure 6). However, the $Cl^-$ concentrations gradually increased in the upper reaches of the Sanhui River and decreased in the lower reaches. The $Cl^-$ concentrations in the estuaries of the Huangjin River and Sanhui River were close to that in the reservoir area, which was consistent with $Na^+$ concentration. The $Cl^-$ concentrations in the tributaries of the Huangjin River also showed a gradually decreasing trend, which was in accordance with the main stream. The $NO_3^-$ concentrations had no obvious spatial variation whether in the main streams or in the tributaries. Previous studies have shown that artificial pollutants were the main source of $NO_3^-$ concentrations [6], and the above phenomena reflected the randomness of human activities in the water environment. The $SO_4^{2-}$ concentrations decreased gradually from upstream to downstream in the main stream of the Baishi River, while those in the main stream and tributaries of the Sanhui River both increased first and then decreased. By comparison, the $SO_4^{2-}$ concentrations in the Huangjin River and the lower reservoir area of the Ganjing River were low, and that spatial fluctuation was not obvious. The $HCO_3^-$ concentrations decreased gradually from upstream to downstream in the main stream of the Huangjin River, Sanhui River, and Baishi River, which was likely related to the decreasing trend of tributaries or the degassing of carbon dioxide. The $HCO_3^-$ concentrations were generally stable, with an average concentration of about 3532.13 umol/L, and decreased to 3025.45 umol/L after passing the Baishi Reservoir.

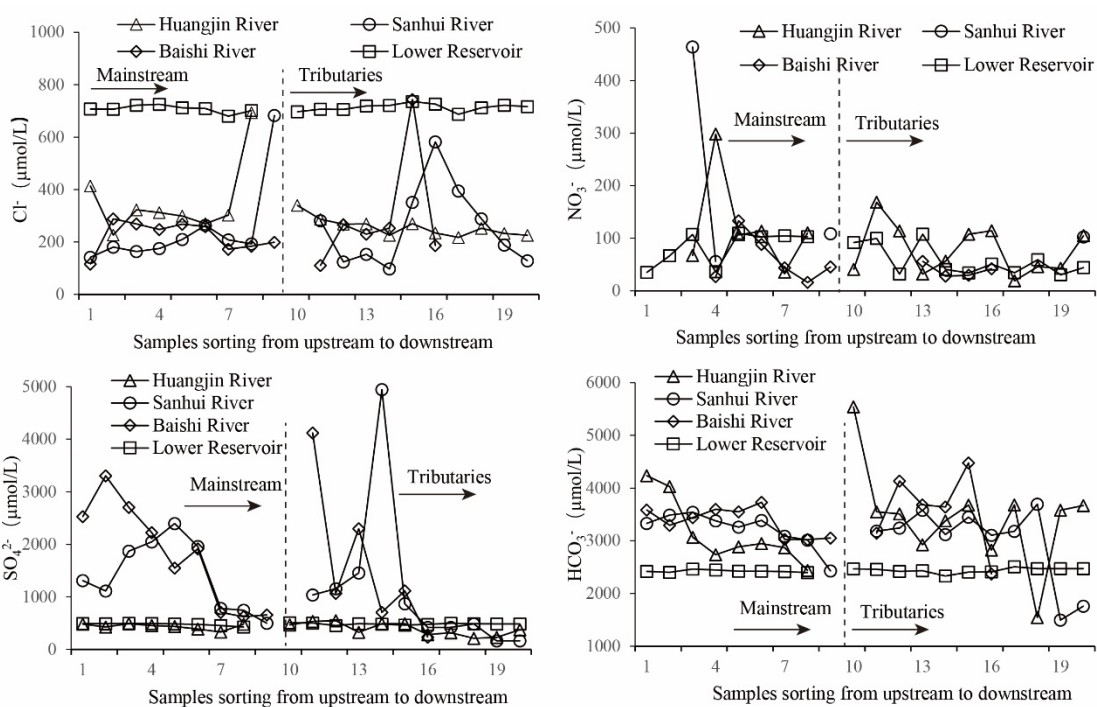

**Figure 6.** The spatial variation of major anions in Ganjing River Basin.

### 4.2. Influence of Rock Weathering on Hydrochemistry

The riverine hydrochemistry was mainly controlled by rock weathering, atmospheric rainfall, and evaporation crystallization under natural conditions [8,11,24]. There were low TDS values and high ratios of $Na^+/(Na^+ + Ca^{2+})$ and $Cl^-/(Cl^- + HCO_3^-)$ in the riverine water controlled by atmospheric rainfall, while high TDS values and low ratios of $Na^+/(Na^+ + Ca^{2+})$ and $Cl^-/(Cl^- + HCO_3^-)$ in the riverine water were controlled by evaporation crystallization. In contrast, the ratios of $Na^+/(Na^+ + Ca^{2+})$ and $Cl^-/(Cl^- + HCO_3^-)$ in the riverine water controlled by rock weathering were generally low [8]. The TDS values in the Ganjing River Basin ranged from 213.00 to 994.33 mg/L (Figure 7), and the ratios of $Na^+/(Na^+ + Ca^{2+})$ and $Cl^-/(Cl^- + HCO_3^-)$ were 0.05~0.46 and 0.03~0.24, respectively, which were generally controlled by rock weathering (Figure 7). According to the distribution of Gibbers, the ratios of $Na^+/(Na^+ + Ca^{2+})$ and $Cl^-/(Cl^- + HCO_3^-)$ were generally high and relatively concentrated in the lower reservoir area of the Ganjing River, compared with other river sections.

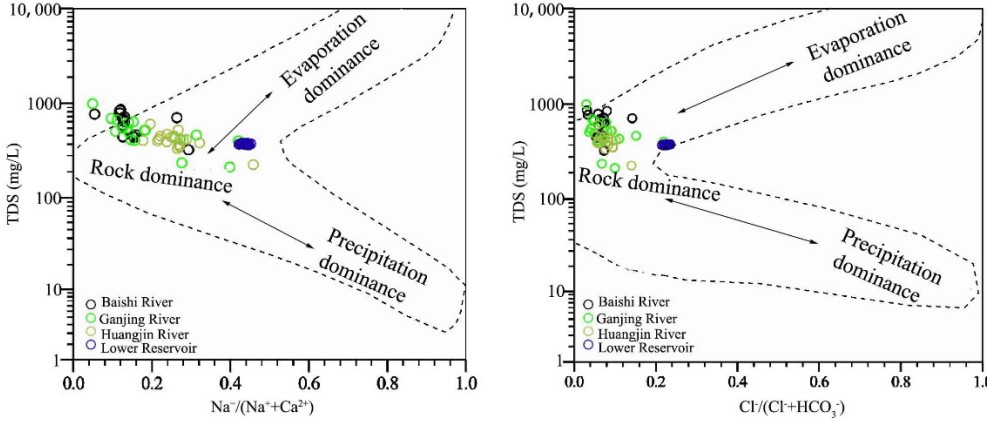

**Figure 7.** The plot of the major ions within the Gibbs.

The concentration ratios of the main ions to $Na^+$ in riverine water were usually standardized to eliminate the influence of the dilution effect and evaporation effect when

the qualitative studies of major ionic sources were conducted [4,11,23]. Results showed that the riverine hydrochemistry characteristics in the basin were located between silicate rock and carbonate rock weathering ends (Figure 8). In contrast, the Baishi River and Sanhui River were more inclined to the end element of carbonate rock weathering, followed by the tributary of the Huanjin River. The riverine hydrochemistry characteristics in the reservoir area were close to the end of silicate rock weathering. The average values of Ca/Na, Mg/Na, and $HCO_3$/Na in the basin were 4.04, 1.21, and 6.21, respectively, which were slightly higher than the average values of hydrochemistry in the Yangtze River Basin (2.51, 5.02, 0.88) [6], much higher than the average values of the global silicate basins (0.35, 0.24, 2) [11], and lower than the Xijiang River Basin, where carbonate rocks were widely exposed (6.14, 1.39, 12.81) [23]. Based on the above analysis, the hydrochemistry characteristics were mainly controlled by the weathering of carbonate rocks and silicate rocks. Although the reservoir was inflowed by several rivers, the ratios of end element in the lower reservoir area of the Ganjing River were lower than the average of above rivers, which revealed that the backwater of the Yangtze River had an important impact on the hydrochemistry characteristics.

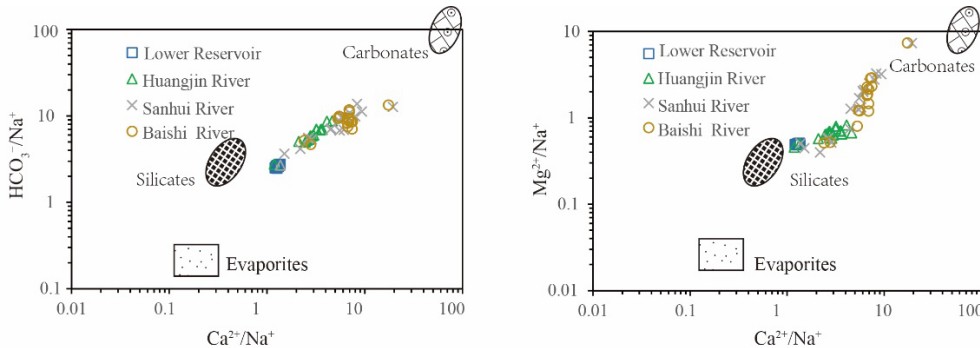

**Figure 8.** Molar ratios of $HCO_3^-$/Na and Ca/Na, and Mg/Na and Ca/Na.

### 4.3. The Estimation of the Ion Sources

According to the above analysis, the ions mainly came from silicate rock and carbonate rock weathering in the Ganjing River. In addition, atmospheric rainfall and human activities were also important sources [25]. The mass balance equation of riverine ions forward modeling could be expressed as:

$$[X]_{riv} = [X]_{atm} + [X]_{car} + [X]_{sil} + [X]_{anth} \tag{1}$$

where $[X]_{riv}$ was the riverine ions; the $[X]_{atm}$, $[X]_{car}$, $[X]_{sil}$, and $[X]_{anth}$ represented the ions from atmospheric input, carbonates weathering, silicates weathering, and anthropogenic inputs, respectively.

#### 4.3.1. Atmospheric Input

The riverine water in the Ganjing River was mainly supplied by atmospheric rainfall, which was an important source of ions in the river [4,26,27]. Atmospheric rainfall was collected in the upper, middle, and lower reaches of the basin during the sampling period, and the ionic compositions were analyzed. The $K^+$, $Na^+$, $Ca^{2+}$, $Mg^{2+}$, $Cl^-$, $NO_3^-$, and $SO_4^{2-}$ were 12.19 μmol/L, 25.93 μmol/L, 23.71 μmol/L, 5.42 μmol/L, 45.89 μmol/L, 6.38 μmol/L, and 22.05 μmol/L, respectively. The ion concentrations sourced from atmospheric input can be calculated according to Equation (2). The annual average rainfall in the Ganjing River Basin was 1200 mm, and the annual runoff depth was 529.67 mm. By calculation, the amount of $K^+$, $Na^+$, $Ca^{2+}$, $Mg^{2+}$, $Cl^-$, $NO_3^-$, and $SO_4^{2-}$ from atmospheric

rainfall accounted for 51.4%, 11.7%, 3.4%, 2.8%, 38.1%, 27.8%, and 11.4% of the riverine ion concentrations, respectively (Table 2).

$$X_{riv}^* = \frac{P}{D} \times X_{rain} \tag{2}$$

where $X_{riv}^*$ was $X$ ionic concentration from atmospheric rainfall in rivers (µmol/L). $X_{rain}$ was the $X$ ionic concentration in atmospheric rainfall. $P$ and $D$ denote the annual average precipitation (1200 mm) and runoff depth (529.67 mm), respectively.

**Table 2.** The contribution rate statistics of ionic sources for TDS in different sections.

| Section | | Atmospheric Input (%) | Agricultural Fertilization (%) | Domestic Water Discharge (%) | Silicate Weathering (%) | Carbonates Weathering (%) |
|---|---|---|---|---|---|---|
| Lower Reservoir Area | Average | 4.66 | 4.68 | 23.11 | 20.12 | 47.47 |
| | Range | 4.57~4.79 | 3.87~5.41 | 21.28~24.29 | 18.68~21.37 | 46.11~50 |
| | Standard Deviation | 0.07 | 0.48 | 0.84 | 0.90 | 1.10 |
| Huangjin Rive | Average | 4.32 | 1.46 | 6.66 | 16.73 | 70.86 |
| | Range | 2.67~8.27 | 0.16~4.50 | 2.57~21.93 | 10.64~33.18 | 49.45~81.11 |
| | Standard Deviation | 1.11 | 1.24 | 4.32 | 4.96 | 8.54 |
| Sanhui River | Average | 3.81 | 0.98 | 4.73 | 11.21 | 79.46 |
| | Range | 1.57~8.34 | 0.02~4.37 | 0.08~21.38 | 2.42~31.51 | 50.38~95.83 |
| | Standard Deviation | 1.69 | 1.09 | 5.15 | 7.27 | 12.54 |
| Baishi River | Average | 3.07 | 1.04 | 4.12 | 8.12 | 83.71 |
| | Range | 1.77~6.01 | 0.01~3% | 0.06~14.13 | 2.50~21.67 | 67.15~95.77 |
| | Standard Deviation | 1.14 | 0.82 | 3.31 | 4.47 | 7.39 |
| The Basin | Average | 4.01 | 2.09 | 9.96 | 14.38 | 69.63 |
| | Deviation | 1.28 | 1.83 | 8.81 | 6.71 | 16.40 |

### 4.3.2. Anthropogenic Input

Previous studies have shown that domestic water discharge and agricultural fertilization were usually the main forms of human activities, which largely caused the enrichment of $Cl^-$, $Na^+$, $K^+$, and $NO_3^-$ in the river [4,6]. There was no evaporite exposed in the Ganjing River Basin, the remaining $Cl^-$ and $NO_3^-$ were all attributed to anthropogenic sources after deducting the atmospheric sedimentation sources. Previous research gave the ratios of $Cl^-$, $NO_3^-$, and $SO_4^{2-}$ between the above material sources, which were 8.5, 1.8, and 2.0, respectively [26]. The ratio of $Cl^-$ to $Na^+$ in the domestic water was close to 1.15 in the sea salt [11]. Taking the molar ratios of $NO_3/Na$ (4), $Cl/Na$ (5), and $K/NO_3$ (0.35) from agricultural activities and $K/Na$ (0.16), $Mg/Na$ (0.2), and $Ca/Na$ (0.8) from urban activities into account [6], the amount of major ions from domestic emission were calculated. Accordingly, the amount of $K^+$, $Na^+$, $Ca^{2+}$, $Mg^{2+}$, $Cl^-$, $NO_3^-$, and $SO_4^{2-}$ from human activities accounted for 17.23%, 31.85%, 12.81%, 9.89%, 61.95%, 72.17%, and 26.26%, respectively. Among the above anthropogenic sources, the ratio of domestic emission sources to agricultural activity sources was 4.79.

### 4.3.3. Rock Weathering Sources

The remaining ions, deducting atmospheric inputs and anthropogenic inputs, were mainly derived from the chemical weathering of carbonate and silicate minerals, which was verified by the end members graph (Figure 8). The ratios of $Mg/Na$ and $Ca/Na$ in the typical silicate and carbonate basins, respectively, were 0.2, 0.35 and 50, 10 [11,28]. According to the above ratios, $K^+$, $Mg^{2+}$, $Ca^{2+}$, and $Na^+$ deducting atmospheric and anthropogenic inputs were divided into silicate and carbonate weathering sources. Correspondingly, the sources of $HCO_3^-$ were divided, considering the equilibrium relationship with cations. According to the stratigraphic setting in the basin, there was not enough evidence to support the existence of pyrite and gypsum. The precipitation (acid precipitation) was likely the

main source of $SO_4^{2-}$, which also participated in rock weathering. Therefore, the remaining ion charges were supplemented by $SO_4^{2-}$, and the $SO_4^{2-}$ produced from rock weathering sources could be calculated.

### 4.3.4. The Contribution Rates

According to the above ionic sources analysis, the mean contribution rates of atmospheric rainfall, agricultural activities, domestic emission, silicate weathering source, and carbonate weathering source in the TDS were 4.01%, 2.09%, 9.96%, 14.38%, and 69.63% (Table 2, Figure 9), respectively. The contribution rates confirmed that the riverine hydrochemistry was mainly controlled by carbonate weathering in the Ganjing River Basin, which were generally consistent with the silicate weathering (12%) and carbonate weathering (63.9%) contribution rates in the Jialing River Basin [29]. The standard deviation of contribution rates of atmospheric rainfall in different regions was ±1.28%, and the deviation was even as low as 0.07% in the reservoir area, which indicated that the contribution rate of atmospheric rainfall in the basin was relatively stable. Anthropogenic inputs contributed 27.80% of ions in the lower reservoir area of the Ganjing River, which was seven times the average value in the other regions of the basin. Previous studies had shown that anthropogenic activities brought 20~30% of dissolved solid in the Chongqing section of the TGR [6], which confirmed that that contribution rate of anthropogenic activities was reasonable, and the hydrochemical characteristics in the lower reservoir area of the Ganjing River were mainly affected by the backwater of TGR. The average contribution rate of rock weathering in the Huangjin River, Sanhui River, and Baishi River Basins was more than 90%, and the contribution rate of carbonates weathering was even up to 78.01%, which illustrated that rock weathering was the main source of dissolved matter in the basin. The average contribution rate of carbonates weathering was 10 times that of silicate weathering in the Baishi River Basin, and the ratios were 7.09, 4.24, and 2.36 in the Sanhui River, Huangjin River, and Lower Reservoir, respectively. The average contribution rate of carbonates weathering was 47.47% in the lower reservoir area of the Ganjing River, which was consistent with that (40~50%) in the Chongqing section of the TGR [6].

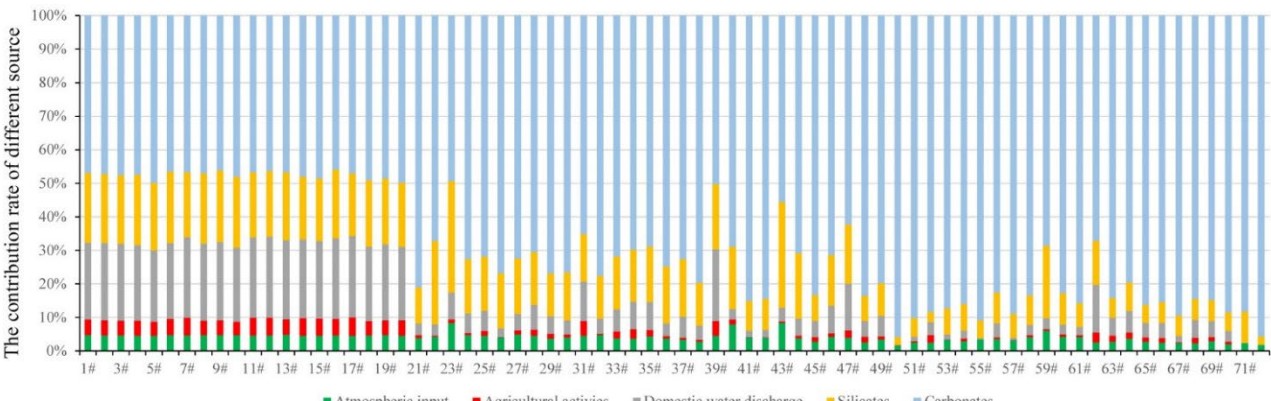

**Figure 9.** The contribution rate statistics of ionic sources for TDS in different sections.

### 4.4. Rock Weathering and $CO_2$ Consumption

The annual average discharge was $4.82 \times 10^8$ m$^3$ in the Ganjing River Basin, of which, 79.6% occurred in the flood season from May to October and 20.4% in the dry season from November to April. Studies have shown that the ionic concentration was the typical power function of riverine runoff (Equation (3)), which was widely used in major rivers in the world [1,29,30]. Accordingly, the ratios of ion concentration in different periods were calculated by the following formula (Equation (4)).

$$C_i = a \times Q^b \tag{3}$$

$$\frac{C_{i_j}}{C_{i_k}} = \left(\frac{Q_j}{Q_k}\right)^b \tag{4}$$

where $C_i$ was the ionic concentration and $Q$ was the discharge in the basin; the $a$ and $b$ denoted the regression parameters of the function. The $j$ and $k$ represented different periods.

Scholars have established the power function relationships between ion concentrations and riverine discharge based on the measured data in the Jialing River Basin [29]. Taking that the contribution of silicate weathering source and carbonate weathering (14.38% and 69.63%) in this basin were similar to those (12% and 63.9%) in the Jialing River Basin into consideration, the riverine ion concentrations under a different discharge were simulated through the above power function relationship (Figure 10), so as to eliminate the dilution effect caused by the change of discharge in the flood season.

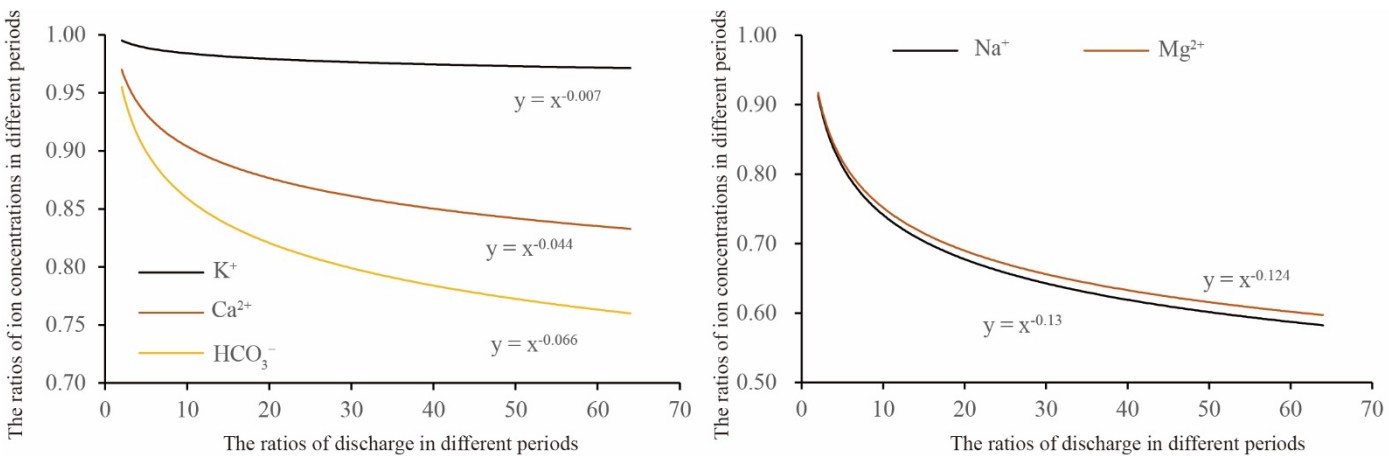

**Figure 10.** Power function relationships between ion concentration ratios and discharge ratios according to previous research [29].

According to the main cation concentrations from silicates and carbonates weathering calculated in Section 4.3, the riverine cation concentrations under different discharges were simulated with the above power function curves (Figure 10), and the cation output fluxes of different sources were counted monthly (Figure 11). Generally, the rock weathering rates including silicates and carbonates were estimated based on the cation fluxes as follows (Equations (5) and (6)).

$$SWR = \Phi K_{sil}^+ + \Phi Na_{sil}^+ + \Phi Ca_{sil}^{2+} + \Phi Mg_{sil}^{2+} \tag{5}$$

$$CWR = \Phi Ca_{carb}^{2+} + \Phi Mg_{carb}^{2+} \tag{6}$$

where $SWR$ and $CWR$ were silicate and carbonate weathering rates (t/km$^2$), and the sil and carb denote the ions from silicates and carbonates weathering. The $\Phi$ meant the monthly fluxes of riverine ions.

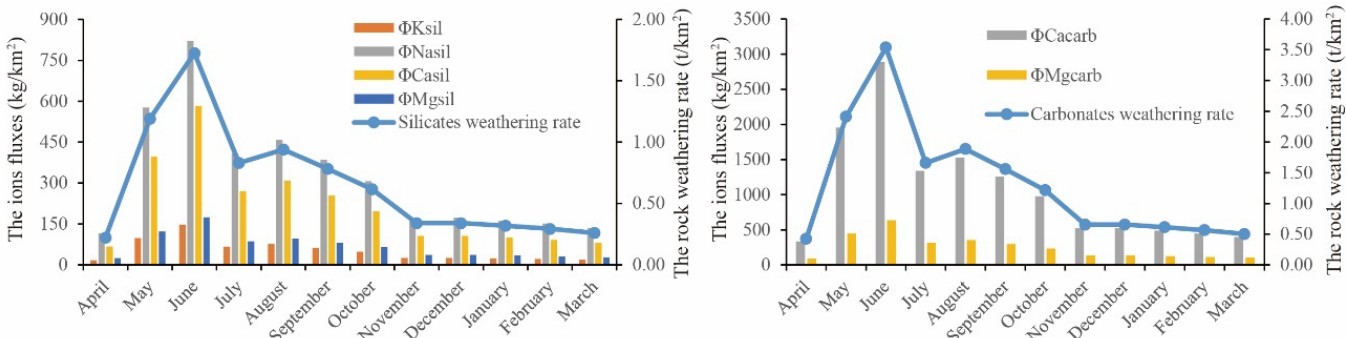

**Figure 11.** The monthly cation output fluxes of different sources and weathering rates.

Accordingly, the rock weathering rate was calculated as 23.54 t/km$^2$ in the basin above 19#, in order to avoid the backwater of the TGR, of which, the silicate and carbonate weathering rates were 7.85 t/km$^2$ and 15.68 t/km$^2$, respectively. Among the existing research results, Chetelat et al. had calculated the silicate and carbonate weathering rates, ranging from 0.7 to 7.1 t/km$^2$/y and 6 to 21 t/km$^2$/y, respectively. Meanwhile, the rock weathering rates were 23.6 t/km$^2$/y in the Jialing River [29] and 22.89 t/km$^2$/y in the Xiaojiang River [31], which were the important tributaries in the TGR. Therefore, the above research results revealed the rationality of rock weathering rates in the Ganjing River. Moreover, the silicate and carbonate weathering rates significantly ($p < 0.01$) correlated with $HCO_3^-$ concentrations (Figure 12). The $H_2CO_3$ formed by $CO_2$ and $H_2O$ in the atmosphere or soil was the important acidic medium involved in rock weathering. On the one hand, the above correlation relationship further revealed the rationality of rock weathering rates in the basin; on the other hand, the rock chemical weathering was the important carbon sink process in the basin.

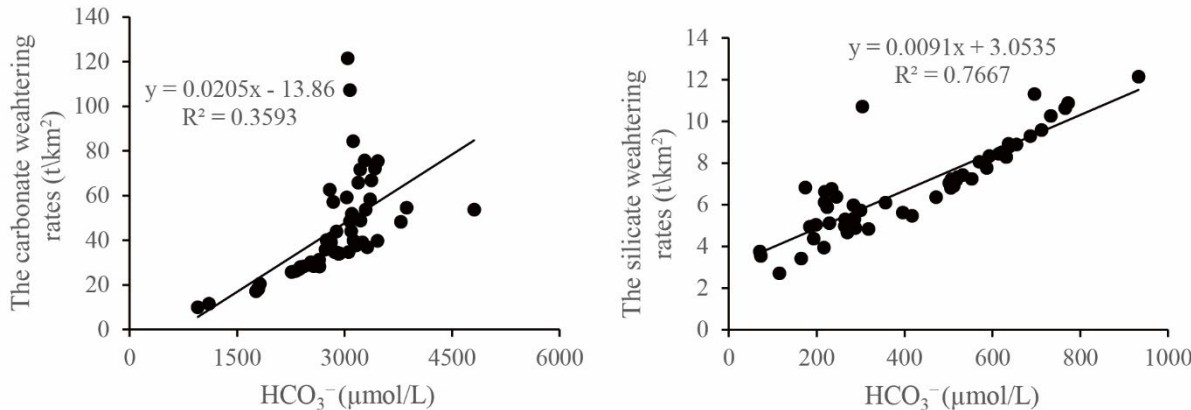

**Figure 12.** The relationship between rock (silicate and carbonate) weathering rates and $HCO_3^-$ concentrations.

Riverine $HCO_3^-$ concentration was the important basis for estimating the carbon sink capacity of rock chemical weathering in the basin, which mainly came from carbonate and the atmospheric or soil $CO_2$ absorbed by rock weathering. Therefore, the half of $HCO_3^-$ stemming from carbonate mineral weathering should be deducted when the actual carbon sink capacity is calculated in the basin [13]. The $HCO_3^-$* monthly output fluxes were simulated by the power function between $HCO_3^-$ concentration after deduction and riverine discharge. On this basis, the $HCO_3^-$* flux produced from silicate and carbonate weathering was calculated by the following Equation (7).

$$HCO_3^{-*} = \left[HCO_3^-\right]_{sil} + \frac{1}{2}\left[HCO_3^-\right]_{carb} \tag{7}$$

$$[\Phi CO_2] = \Phi HCO_3^{-*}/S \tag{8}$$

Accordingly, the atmospheric $CO_2$ consumption flux was $6.88 \times 10^5$ mol/y/km$^2$ (Equation (8)), of which, the $2.67 \times 10^5$ mol/y/km$^2$ and $3.59 \times 10^5$ mol/y/km$^2$ originated from silicates and carbonates weathering, respectively. The annual $CO_2$ flux absorbed by rock weathering in the Ganjing River Basin was $0.63 \times 10^9$ mol/y, accounting for about 0.75‰ of the $CO_2$ flux ($837 \times 10^9$ mol/y) above Datong Station in the Yangtze River Basin; however, its controlling area only accounted for 0.50‰ of the Yangtze River Basin [6], which revealed that there was high carbon sink potential in the Ganjing River Basin.

## 5. Conclusions

The Ganjing River located in the hinterland of TRG was systematically investigated to analyze the ionic composition and spatial variation, quantify the ionic source and influencing factors by the forward model, and assess the chemical weathering rates and $CO_2$ consumption. The following conclusions were as follows:

(1) The water showed moderately alkalescent in the Ganjing River Basin, with the pH value ranging from 7.87 to 8.64. The riverine physical and chemical parameters showed obvious spatial distribution characteristics, which were relatively stable in the lower reservoir area of the Ganjing River. The TDS value fluctuated from 213.00 to 994.33 mg/L, with an average value of $473.31 \pm 154.87$ mg/L, which was higher than that of the global rivers' average.

(2) The hydrochemistry was controlled by the type of "$HCO_3^-$–$Ca^{2+}$". The $Ca^{2+}$ was the main component in the cations and accounted for 43.83~74.53% of the total anions. The $HCO_3^-$ ranged from 1542.87 to 5542.16 μmol/L and accounted for 70.48% of total anions. Most ions except $NO_3^-$ were relatively stable in the lower reservoir area of the Ganjing River, where the concentrations of $Ca^{2+}$ and $HCO_3^-$ were obviously lower than those in other river sections. The above phenomena were largely due to the water mixing caused by the backflow of the reservoir area.

(3) Rock weathering was the main source of riverine ions in the basin, accounting for 84.01% of the total ions, of which, the silicate weathering source and carbonate weathering source contributed 14.38% and 69.63%, respectively. The carbonate weathering generally dominated the hydrochemical characteristics, which was also demonstrated by the analysis of ions' components. The contribution rates of atmospheric rainfall were relatively low and stable in different river sections, with an average of $4.01 \pm 1.28$%. Anthropogenic activities contributed 12.05% of riverine ions, which were an important source of dissolved matter that cannot be ignored. Especially, anthropogenic inputs contributed 27.80% of ions in the lower reservoir area of the Ganjing River.

(4) Based on the empirical power functions, the rock weathering rate was calculated as 23.54 t/km$^2$ in the Ganjing River Basin, which already eliminated the dilution effect caused by the change of discharge. The silicate and carbonate weathering rates were 7.85 t/km$^2$ and 15.68 t/km$^2$, respectively. Accordingly, the atmospheric $CO_2$ consumption flux was $6.88 \times 10^5$ mol/y/km$^2$, of which, the $2.67 \times 10^5$ mol/y/km$^2$ and $3.59 \times 10^5$ mol/y/km$^2$ originated from silicates and carbonates weathering. The annual $CO_2$ consumption flux accounted for about 0.75‰ of the $CO_2$ flux ($837 \times 10^9$ mol/y) in the Yangtze River Basin, which revealed that there was high carbon sink potential in the Ganjing River Basin.

**Author Contributions:** Conceptualization, L.D.; Formal analysis, C.Z.; Investigation, K.J., W.L., Y.L. and Y.H.; Data curation, R.Z.; Writing—review & editing, Q.Z.; Project administration, L.D.; Funding acquisition, X.G. All authors have read and agreed to the published version of the manuscript.

**Funding:** This research was funded by Knowledge Innovation Program of Wuhan-Shuguang [2022020801020245], Central Public-interest Scientific Institution Basal Research Fund of China [CKSF2021744/TB], Open Project Program of Chongqing Key Laboratory of Karst Environment [Cqk202101], and National Key R&D Program of China [2021YFE0111900].

**Conflicts of Interest:** The authors declare no conflict of interest.

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
