# Peer review of "The Hydrochemistry, Ionic Source, and Chemical Weathering of a Tributary in the Three Gorges Reservoir"

_sustainability, doi:10.3390/su142215376_

Round 1

Reviewer 1 Report

The comments have been attached.

Author Response

  We thank you for the constructive comments to improve our manuscript. The manuscript has been carefully revised on the basis of the comments. All changes in our manuscript are clearly marked in red. The revisions to our manuscript are as follows:

â‘ Question: Equation 2 in section 4.3: Sulfate is one of the most important anions in the study rivers (164-4943 μmol/L) and may account for ~30% of the total anions according to Figure 3. Most sulfate originates from sulfate oxidation (such as pyrite), evaporites dissolution (gypsum), and precipitation (acid precipitation). This means that sulfate oxidation and/or evaporites dissolution may also important chemical weathering processes in the study catchments. The authors should add these two sources’ of discrimination in Equation 2 and the corresponding description.

Answer: We applied the demonstration of sulfate source in order to support our research. Correspondingly, the sources of HCO3- were divided, considering the equilibrium relationship with cation. According to stratigraphic setting in the basin, there were not enough evidence to support the existence of pyrite and gypsum. The precipitation (acid precipitation) was likely the main source of SO42-, which also participated rock weathering. Therefore, the remaining ion charges were supplemented by SO42-, and the SO42- producing from rock weathering source could be calculated. (see the section of 4.3.3. Rock weathering sources).

â‘¡Question: Section 4.4: Ionic concentration of the Ganjing River in other seasons is calculated by the power function relationships between ionic concentration and river discharge obtained in the Jialing Jiang River basin. This method may result in large uncertainties in the calculation of chemical weathering rates because these two basins has totally different lithology. Silicate is the dominant lithology in the Jialing Jiang River basin, while carbonate has much higher coverage in the Ganjing River basin.

Answer: We have just considered the lithology similarity of the two basins, which were also demonstrated by the contribution of silicate weathering source and carbonate weathering in the 4.3.4. The contribution rates. And, we applied the further supplementary such as “Taking that the contribution of silicate weathering source and carbonate weathering (14.38% and 69.63%) in this basin were similarity to those ((12% and 63.9%) in Jialing River basin into consideration, the riverine ion concentrations under different discharge were simulated through the above power function relationship (Figure 11), so as to eliminate the dilution effect caused by the change of discharge in flood season.” in the 4.4. Rock weathering and CO2 consumption.

â‘¢Question: Chemical weathering rate in the study basin is only 23.54 t/km2/y. This value is much lower than the corresponding value in the neighboring basins, such as the Wu Jiang (105 t/km2/y) (Han et al., 2010), Xi Jiang (86 t/km2/y) (Xu and Liu, 2010), and Lower Chang Jiang (38 t/km2/y) (Chetelat et al., 2008). As a comparison, the CO2 consumption rate caused by chemical weathering is as high as 8.92×105 mol/km2/y, which is even comparable with that in the Xi Jiang basin (9.6 mol/km2/y), Wu Jiang basin (9.31 mol/km2/y), and two times higher than the lower Chang Jiang (4.97 mol/km2/y). The authors should double-check the data used for the calculation of chemical weathering rates and CO2 consumption rates in this study.

Answer: Thank the reviewer very much for the reminder of CO2 consumption rate. The half of HCO3- stemming from carbonate mineral weathering should be deducted when the actual carbon sink capacity was calculated in the basin. However, we ignored this and had made a very serious computational error. After correction, the CO2 consumption rate was 6.88×105 mol/y/km2, which was comparable with that in Xiaojiang River (another tributary in the Three Gorges Reservoir Region). See in the section of 4.4. Rock weathering and CO2 consumption.

â‘¢Question: 1. This manuscript did not add line numbers. Thus it is not easy for the reviewer(s) to

make a comment or suggestion on some specific sentences.

Answer: We have added the line numbers in the paper.

â‘£Question: Language should be polished by native speakers because some words or sentences

in this manuscript are quite confusing.

Answer: We carefully readed the full text and made modifications word by word, which was marked in red colour.

⑤Question: The lithology map of the study basin should be added to this manuscript.

Answer: We modified the Figure1 and added lithology information.

â‘¥Question: Figure 3: dots in this photo is hard to discriminate for the different tributaries. It it

recommended that different colors be used to make a distinction.

Answer: According to the suggestions, we redraw the Figure3.

 â‘¦Question: Section 3.3: what is the RES?

Answer: The RES is resistivity, which was described in the 2.2. Sampling section.

⑧Question: Section 4.1.1: “the above phenomenon were likely due to ...in the reservoir”. This

sentence is quite confusing. It is hard for the readers to follow the authors’ explanations.

Answer: We revised the sentence as “The above phenomena were likely due to the water mixed effect and stratification in the lower reservoir.”  in the Section 4.1.1.

⑨Question: The word “Gibbers” in title of Figure 8 should be replaced by “Gibbs”.

Answer: We revised it.

â‘©Question: The first sentence of section 4.2: I am not sure if the atmosphere rainfall is a

geochemical process. The description in this manuscript should be more concise.

Answer: We revised the description “The riverine hydrochemistry was mainly controlled by rock weathering, atmospheric rainfall and evaporation crystallization under natural conditions.”

⑪Question: 10. Section 4.3.4: The authors did not introduce the methods used to discriminate the contribution from agricultural activities and domestic emissions in Equation 2 and section 4.3.2.

Answer: The contribution from agricultural activities and domestic emissions were calculated according to “The previous research had given the ratios of Cl-, NO3- and SO42- between above material source, which were 8.5, 1.8 and 2.0, respectively (Li, et al., 2009). The ratio of Cl- to Na+ in the domestic water was close to 1.15 in the sea salt(Gaillardet, et al., 1999). Taking the molar ratios of NO3/Na (4), Cl/Na (5) and K/NO3 (0.35) from agricultural activities and K/Na (0.16), Mg/Na (0.2) and Ca/Na (0.8) from urban activities into account(Chetelat, et al., 2008), the amount of major ions from domestic emission were calculated.” See in the section 4.3.2.

â‘«Question: Section 4.3.1: the data of annual average precipitation and runoff depth used in Equation 2 is 1508 mm and 810 mm, respectively. However, these two value in the study basin is about 1200 mm and 530 mm, respectively. Why the authors did not use the data observed in the study regions?

Answer: We thank expert very much for finding our error. We corrected the above data “P and D denoted the annual average precipitation (1200 mm) and runoff depth of (529.67mm), respectively.”

⑬Question: The discrimination about the contribution of silicate and carbonate weathering to riverine major cations did not be described in detail even though this procedure matters the calculation of chemical weathering rates and carbon sink rates.

Answer: The contribution of silicate and carbonate weathering to riverine major cations was calculated according to “The remaining ions deducting atmospheric input and anthropogenic inputs mainly derived from chemical weathering of carbonate and silicate minerals, which was verified by the end members graph (Figure 9). The ratios of Mg/Na and Ca/Na in the typical silicate and carbonate basins respectively were 0.2, 0.35 and 50, 10(Gaillardet, et al., 1999; Han, et al., 2004). According to the above ratios, K+, Mg2+, Ca2+ and Na+ deducting atmospheric and anthropogenic inputs were divided into silicate and carbonate weathering sources.” in the 4.3.3. Rock weathering sources section.

â‘­Question: Figure 10: the label in this cartoon seems wrong: the light blue column should be carbonate dissolution while golden is silicate weathering.

Answer: We redrawed the Figure 10 and corrected relevant information.

â‘®Question: Figure 1 and Figure 3 in the paper and the text descriptions on the figures are too small.

Answer: We redrawed the Figure 1 and Figure 3.

⑯Question: the contribution rates (page 15), it is ambiguous that the author argued "The average contribution rate of carbonates weathering was 10 times of silicate weathering in Baishi River basin, the ratios were respectively 7.09, 4.24 and 2.36.

Answer: We revised the sentence “The average contribution rate of carbonates weathering was 10 times of silicate weathering in Baishi River basin, and the ratios were 7.09, 4.24 and 2.36 in Sanhui River, Huangjin River and Lower Reservoir, respectively.

â‘°Question: References are not in alphabetical order, and more similar literature should be

cited in this manuscript.

Answer: We have revised the references and supplemented the relevant literature.

Reviewer 2 Report

The comments on the paper entitled “The Hydrochemistry, Ions Source and Chemical Weathering of Tributary in the Three Gorges Reservoir” (sustainability-2014274). This paper investigated the spatial variation of riverine ions and their sources, and estimated the chemical weathering rates and CO2 consumption in the typical tributary of the Three Gorges Reservoir. Overall, this is an interesting work because the this work could provide new knowledge and data serve for the integrated management water resources in the Ganjing River basin under the influence of Three Gorges Reservoir impounding. Thus, according to my own judgement and knowledge, I recommend accepting it for publication after minor corrections. Reasons of my recommendations are followings:c

Major comments:

1) The language of this manuscript needs to be improved. There are numbers of careless typos and grammatical errors that hinder to understand authors' discussion. They should check their manuscript more carefully or send to English-proofing before sending to review process.

2) The abstract should reflect your findings more clearly while it gives the readers a good idea of what they are going to read through the paper. It requires more precise wording.

3) In the Introduction section: The introduction didn’t clearly summary the research progress of river chemical weathering rates and CO2 consumption, especially in the reservoir area. Please consult some relevant papers to modify this section! Furthermore, the gap and contribution of this paper were not well presented in the last paragraphs of the Introduction.

4) In the “Study Area and Methods” section: The first paragraph of this section described the information of study area. So, I recommend dividing this section into three subsections: 2.1. Study area, 2.2. Sampling, and 2.3. Methods and data.

5) In the “3. Results” section: Generally, in order to comparison the riverine ion concentrations with those in other rivers, the concentration unit of water main ions is mg/L. Moreover, the right plot in Figure 3 was not standard. Please refer to the Piper diagram in this article: Wang, P., Yu, J.J., Zhang, Y.C., Liu, C.M., 2013. Groundwater recharge and hydrogeochemical evolution in the Ejina Basin, northwest China. J. hydrol. 476, 72–86.

6) The text in the “4.1. Spatial differentiation of hydrochemical characteristics” section mainly described the results in Figures 5, 6 and 7, with little discussion. So, I suggested moving these section into the “3. Results” section.

7) In the “4.3.4. The contribution rates” section (Page 14-16): Which hydrochemical parameters were the contribution rate in Figure 10 and Table 2 were for? For example, TDS? Please state clearly. Furthermore, multiple sampling of the same sampling point is needed to determine the contribution rate. Based on Through a sampling, what method was used to calculate the contribution rate of this hydrochemical parameter? Please supplement the calculation formula, and describe it detailed.

Minor comments:

1) Figure 1: Latitude and longitude were not marked in Figure 1, please add.

2) Figures 5, 6 and 7: What’s the meaning of “Samples from upstream to downstream” in X axis? It represent the sample sites from upstream to downstream, or the distance? Please describe it clearly.

3) Table 1: Please fix the typo of “NO3”, “SO4” and “HCO3” into “NO3”, “SO4”, “HCO3”, respectively.

4) The unit of the “conductivity (Cond)” is μS/cm. Please change all the "μs/cm" into "μS/cm".

Author Response

  We thank you for the constructive comments to improve our manuscript. The manuscript has been carefully revised on the basis of the comments. All changes in our manuscript are clearly marked in red. The revisions to our manuscript are as follows:

â‘ Question: The language of this manuscript needs to be improved. There are numbers of careless typos and grammatical errors that hinder to understand authors' discussion. They should check their manuscript more carefully or send to English-proofing before sending to review process.

Answer: We had seriously optimized the language of the current manuscript. The modified contents were marked in red.

â‘¡Question: The abstract should reflect your findings more clearly while it gives the readers a good idea of what they are going to read through the paper. It requires more precise wording.

Answer: We have carefully revised the abstract of the article and refined the language expression, such as “Riverine dissolved matter reflects the geochemical genesis information, which is vital to understand and manage water environment. The results showed that the total dissolved solid value (473.31 ± 154.87 mg/L) with the type of “HCO3-- Ca2+” was higher than the average of global rivers. The hydrochemical parameters were relatively stable in the lower reservoir area of Ganjing River,” (See the red word of Abstract section).

â‘¢Question: In the Introduction section: The introduction didn’t clearly summary the research progress of river chemical weathering rates and CO2 consumption, especially in the reservoir area. Please consult some relevant papers to modify this section! Furthermore, the gap and contribution of this paper were not well presented in the last paragraphs of the Introduction.

Answer: The description that “Since full impoundment stage in 2010, TGR and its tributaries have suffered from various environmental problems, such as soil erosion, increased water level, slower flow velocity, extended water retention and large water temperature fluctuations (Ma, et al., 2016). According to the investigation, about 38 tributaries have experienced water blooms in varying degrees, and the water quality in some tributaries bay has exceeded normal standard (Xiang, et al., 2021). The geochemistry characteristics and ions source analysis in major tributaries are significant to grasp evolution process of water environment in large-scale reservoir ecosystem.” has already the significance of this paper. And We have carefully revised Introduction section and demonstrated the impact of reservoir, such as “At present, the global environment is facing huge risks and challenges (Sun, et al., 2022), human activities are widely affecting the surface environment. According to the latest statistics of International Commission on Large Dams, there are 59000 dams with the height of more than 15 meters or a reservoir capacity of more than 3 ×108 m3 in the world(http://www.icold-cigb.org). The construction of the reservoir has destroyed the connectivity and hydrological conditions, hindered the migration of biogenic elements along the river network, and affected the ecological environment in downstream areas (Maavara, et al., 2020).” (See the red word of Introduction section).

â‘£Question: In the “Study Area and Methods” section: The first paragraph of this section described the information of study area. So, I recommend dividing this section into three subsections: 2.1. Study area, 2.2. Sampling, and 2.3. Methods and data.

Answer: According to the suggestions, we have divided the “Study Area and Methods” section into three subsections: 2.1. Study area, 2.2. Sampling, and 2.3. Methods and data. (see the section of 2. Study Area and Methods)

⑤Question: In the “3. Results” section: Generally, in order to comparison the riverine ion concentrations with those in other rivers, the concentration unit of water main ions is mg/L. Moreover, the right plot in Figure 3 was not standard. Please refer to the Piper diagram in this article: Wang, P., Yu, J.J., Zhang, Y.C., Liu, C.M., 2013. Groundwater recharge and hydrogeochemical evolution in the Ejina Basin, northwest China. J. hydrol. 476, 72–86.

Answer: When comparing with other rivers, we have noticed the corresponding unit conversion of main ions. We have cited the article “Wang, P., Yu, J.J., Zhang, Y.C., Liu, C.M., 2013. Groundwater recharge and hydrogeochemical evolution in the Ejina Basin, northwest China. J. hydrol. 476, 72–86.” and modified the Figure 3

â‘¥Question: The text in the “4.1. Spatial differentiation of hydrochemical characteristics” section mainly described the results in Figures 5, 6 and 7, with little discussion. So, I suggested moving these section into the “3. Results” section.

Answer: Considering that the “3. Results” section is a general description of the hydrochemical parameters, “4.1. Spatial differentiation of hydrochemical characteristics” section focuses on the spatial differentiation and influencing factors. The two parts have different emphasis on the content, so we think that the 4.1 part is more appropriate. If the experts still think it is necessary to adjust, we will make further modifications in the future.

⑦Question: In the “4.3.4. The contribution rates” section (Page 14-16): Which hydrochemical parameters were the contribution rate in Figure 10 and Table 2 were for? For example, TDS? Please state clearly. Furthermore, multiple sampling of the same sampling point is needed to determine the contribution rate. Based on Through a sampling, what method was used to calculate the contribution rate of this hydrochemical parameter? Please supplement the calculation formula, and describe it detailed.

Answer: The contribution rate in Figure 10 and Table 2 were for TDS which was cleared in the “4.3.4. The contribution rates” section. The Ganjing River located in the hinterland of The Three Gorges Reservoir was systematically investigated to quantify the ionic source by the forward model(Equation (1))

⑧Question:Figure 1: Latitude and longitude were not marked in Figure 1, please add.

 Answer: We redrawed the Figure 1 and added the latitude and longitude. See in the Figure 1.

⑨Question: Figures 5, 6 and 7: What’s the meaning of “Samples from upstream to downstream” in X axis? It represent the sample sites from upstream to downstream, or the distance? Please describe it clearly.

Answer: The X axis means sampling points successively from upstream to downstream, dividing into main stream and branch stream at intervals.We have redrawn the figure and marked it, hoping it will be easier to understand. See in the Figures

â‘©Question:Table 1: Please fix the typo of “NO3”, “SO4” and “HCO3” into “NO3 ”, “SO 4 ”,“HCO 3 ”, respectively. The unit of the “conductivity (Cond)” is μS/cm. Please change all the "μs/cm" into "μS/cm".

Answer: We have fixed the typo of of “NO3”, “SO4” and “HCO3” into “NO3 ”, “SO 4 ”,“HCO 3 ”, respectively. We have changed the unit of the “conductivity (Cond)” as μS/cm in the paper.

Reviewer 3 Report

I have read the manuscript entitled " The Hydrochemistry, Ions Source and Chemical Weathering of Tributary in the Three Gorges Reservoir " submitted to Sustainability.  In general, the paper was well written. However, before its acceptance, the following revisions and comments must be considered. My detailed comments are listed below.

·         Abstract: “The Ganjing River located in the hinterland of The Three Gorges Reservoir was systematically investigated …” Please explain what is meant by "systematically investigated".

·         Abstract: What is the reason for choosing "Ganjing River" as the study area?

·         "average of global rivers" what is this value?

·         Introduction: Please provide information about other parts of the study at the end of the introduction.

·         In the Introduction section, give information about previous similar studies in this basin. Show the differences of your work in this section.

·         The quality of Figure 1 needs to be improved.

·         2.1. Sampling: “The water environment was systematically monitored in December 2020 along the main tributaries of Ganjing River.”

·         The quality of the figures needs to be improved. Please explain what is meant by " systematically monitored ". How many times were monitoring studies done? What were the time intervals? Can you add a few photos of the monitoring work?

·         How were the locations of the monitoring stations determined?

·         Results: “… was higher than the global rivers’ average (Gaillardet, et al., 1999).” what is this value? "Gaillardet, et al., 1999" is very outdated. Is it correct to compare the data with this study?

·         I couldn't understand Figure 4. It should be explained in detail why this figure was given. The quality of Figure 4 needs to be improved.

·         Figures 5, 6 and 7 show a lack of data. What is the reason?

·         4.3.1 Atmospheric input: “…basin during the sampling period and hydrochemistry analysis were conducted” sampling period?

·         Conclusions: “…which was higher than the average of global rivers.” average of global rivers?

·         In the conclusion part, the purpose of the study should be given again.

·         Flow chart of modeling studies should be presented.

·        The quality of the figures needs to be improved

Author Response

  We thank you for the constructive comments to improve our manuscript. The manuscript has been carefully revised on the basis of the comments. All changes in our manuscript are clearly marked in red. The revisions to our manuscript are as follows:

â‘ Question: abstract: “The Ganjing River located in the hinterland of The Three Gorges Reservoir was systematically investigated …” Please explain what is meant by "systematically investigated".

Answer: We carried out systematic monitoring in the main trunk and tributaries from the downstream to the upstream. 72 samples were collected for testing ions concentration. And the water temperature (Temp), pH, dissolved oxygen (DO), conductivity (Cond), total dissolved solid (TDS), oxidation-reduction potential (ORP) and resistivity (RES) were measured by AP-7000. See in the section of 2.2. Sampling.

â‘¡Question: Abstract: What is the reason for choosing "Ganjing River" as the study area.

Answer: Located in the hinterland of TGR, the Ganjing River is the vital tributary to support the economic and social development in Zhongxian County. About 8 km back-water area is formed due to the impoundment of TGR, which affects the water quality and hydrology in Ganjing River basin. In fact, the water deterioration, eutrophication and algal blooms are seriously restricting the ecological security of the basin. See in the section of 1. Introduction

â‘¢Question: · "average of global rivers" what is this value?

Answer: We revised the description that “The results showed that the total dissolved solid value (473.31 ± 154.87 mg/L) with the type of “HCO3-- Ca2+” was higher than that of global rivers’ average.” in the abstract.

â‘£Question: Introduction: Please provide information about other parts of the study at the end of the introduction. In the Introduction section, give information about previous similar studies in this basin. Show the differences of your work in this section.

Answer: The description that “Since full impoundment stage in 2010, TGR and its tributaries have suffered from various environmental problems, such as soil erosion, increased water level, slower flow velocity, extended water retention and large water temperature fluctuations (Ma, et al., 2016). According to the investigation, about 38 tributaries have experienced water blooms in varying degrees, and the water quality in some tributaries bay has exceeded normal standard (Xiang, et al., 2021). The geochemistry characteristics and ions source analysis in major tributaries are significant to grasp evolution process of water environment in large-scale reservoir ecosystem.” has already clarified the significance of this paper. And We have carefully revised Introduction section and demonstrated the impact of reservoir, such as “At present, the global environment is facing huge risks and challenges (Sun, et al., 2022), human activities are widely affecting the surface environment. According to the latest statistics of International Commission on Large Dams, there are 59000 dams with the height of more than 15 meters or a reservoir capacity of more than 3 ×108 m3 in the world(http://www.icold-cigb.org). The construction of the reservoir has destroyed the connectivity and hydrological conditions, hindered the migration of biogenic elements along the river network, and affected the ecological environment in downstream areas (Maavara, et al., 2020).” (See the red word of Introduction section).

⑤Question: The quality of Figure 1 needs to be improved.

Answer: We redrawed the Figure 1.

â‘¥Question: Please explain what is meant by " systematically monitored ". How many times were monitoring studies done? What were the time intervals? Can you add a few photos of the monitoring work?

Answer: We carried out systematic monitoring in the main trunk and tributaries from the downstream to the upstream in December 24 to29 2020. 72 samples were collected for testing ions concentration. And the water temperature (Temp), pH, dissolved oxygen (DO), conductivity (Cond), total dissolved solid (TDS), oxidation-reduction potential (ORP) and resistivity (RES) were measured by AP-7000. See in the section of 2.2. Sampling. Considering the length of this article, we no longer placed work photos on-site in the paper. If this is needed, we will make a supplement later.

⑦Question:·How were the locations of the monitoring stations determined?

Answer: We carried out systematic monitoring in the main trunk and tributaries at the confluence point of main trunk and tributaries.

⑧Question: I couldn't understand Figure 4. It should be explained in detail why this figure was given. The quality of Figure 4 needs to be improved.

Answer: Considering that Figure 4 does not play a key supporting role in the text, we deleted the figure based on the opinions of experts.

⑨Question: Figures 5, 6 and 7 show a lack of data. What is the reason?

Answer: The break points of polyline in the Figures 5, 6 and 7 are the boundary between the main stream and the tributary, not the lack of data. We have redrawn the figure and marked it, hoping it will be easier to understand. See in the Figures 5, 6 and 7

â‘©Question: 4.3.1 Atmospheric input: “…basin during the sampling period and hydrochemistry analysis were conducted” sampling period?

Answer: Atmospheric rainfall was collected in the upper, middle and lower reaches of the basin during the sampling period, and the ionic compositions were analyzed. The K+, Na+, Ca2+, Mg2+, Cl-, NO3- and SO42- were 12.19µmol/L, 25.93µmol/L, 23.71µmol/L, 5.42µmol/L, 45.89µmol/L, 6.38µmol/L and 22.05µmol/L, respectively.

⑪Question: Conclusions: “…which was higher than the average of global rivers.” average of global rivers?

Answer: The TDS value fluctuated from 213.00 to 994.33mg/L, with an average value of 473.31 ± 154.87 mg/L, which was higher than that of global rivers’ average.

â‘«Question: In the conclusion part, the purpose of the study should be given again.

Answer: The purpose of the study was supplied in the conclusion part. “The Ganjing River located in the hinterland of The Three Gorges Reservoir was systematically investigated to analyse the ions composition and spatial variation, quantify the ionic source and influencing factors by the forward model, and assess the chemical weathering rates and CO2 consumption. The following conclusions were as follows:

⑬Question: The quality of the figures needs to be improved

Answer: The quality of the figures was improved.

Round 2

Reviewer 1 Report

The modifications made by the authors in this manuscript have met the requirements proposed by the reviewer.

Reviewer 3 Report

The authors have responded to all the comments provided by reviewers and me satisfactorily.  I am please to accept the manuscript